# KB-DMGen: Knowlege-Based Global Guidance and Dynamic Pose Masking for Human Image Generation

## Abstract

Recent methods using diffusion models have made significant progress in Human Image Generation (HIG) with various control signals such as pose priors. In HIG, both accurate human poses and coherent visual quality are crucial for image generation. However, most existing methods mainly focus on pose accuracy while neglecting overall image quality, often improving pose alignment at the cost of image quality. To address this, we propose Knowledge-Based Global Guidance and Dynamic pose Masking for human image Generation (KB-DMGen). The Knowledge Base (KB), implemented as a visual codebook, provides coarse, global guidance based on input text-related visual features, improving pose accuracy while maintaining image quality, while the Dynamic pose Masking (DM) offers fine-grained local control to enhance precise pose accuracy. By injecting KB and DM at different stages of the diffusion process, our framework enhances pose accuracy through both global and local control without compromising image quality. Experiments demonstrate the effectiveness of KB-DMGen, achieving new state-of-the-art results in terms of AP and CAP on the HumanArt dataset. The project page and code will be available.

## 1 Introduction

The goal of human image generation (HIG) is to generate high-quality images under certain conditions based on a series of prompts (e.g., pose (Ju et al., 2023b)). HIG serves a wide range of real-world applications, including animation (Corona et al., 2025), game production (Pan et al., 2024), and other fields.

Previous methods (Men et al., 2020; Ma et al., 2017; Tang et al., 2020) require a source image during training using variational autoencoders (VAEs) (Kingma et al., 2013) or Generative Adversarial Networks (GANs) (Goodfellow et al., 2020) for dictating the style of the generated images. These methods synthesize target images with specific human features by adjusting the source images, but the training process of these methods is unstable and highly dependent on the distribution of the source images. Recent advances in controllable text-to-image (T2I) Stable Diffusion (SD) (Rombach et al., 2022) show potential to eliminate the need for source images, enabling greater creative freedom through reliance on text prompts and external conditions (Zhang et al., 2023; Zhao et al., 2023; Mou et al., 2024; Li et al., 2023). These methods often face challenges in accurately matching conditional images with sparse representations such as skeleton pose data (Ju et al., 2023b). To achieve accurate pose control, various pose guided T2I methods are proposed, such as introducing pose heatmap supervision loss (Ju et al., 2023b), establishing a graph topological structure between the pose priors and latent representation of diffusion models (Yin et al., 2025), applying pose masks to the attention module of the ViT (Wang et al., 2024). These strategies effectively guide the network to focus on pose regions, thereby improving pose fidelity.

While precise pose alignment is essential, high-quality human image generation also demands the guidance of global visual semantics to ensure overall image quality. However, these methods (Yin et al., 2025; Ju et al., 2023b; Wang et al., 2024) emphasize the modeling of pose details while neglecting overall image quality. To address this issue, we propose Knowledge Based Global Guidance and Dynamic pose Masking for HIG (KB-DMGen) in Fig. 1. KB-DMGen introduces a visual

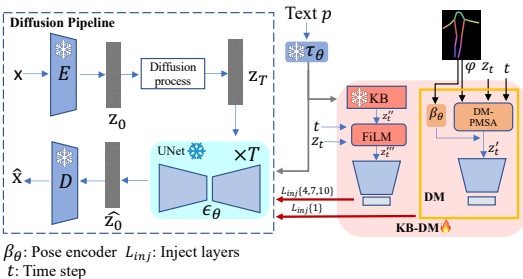

Figure 1: Overview of our method. The visual Knowledge Base (KB) provides global guidance by encoding visual features related to the text description, while the Dynamic pose Mask (DM) enables fine-grained local control. They achieve unified control of both global semantics and local details.

Figure 2: The framework of KB-DMGen. The text encoding is used to retrieve semantic codebook features from a visual Knowledge Base (KB), providing global semantic guidance; meanwhile, pose information generates temporally dynamic masks through the diffusion process, enabling precise control over human pose.

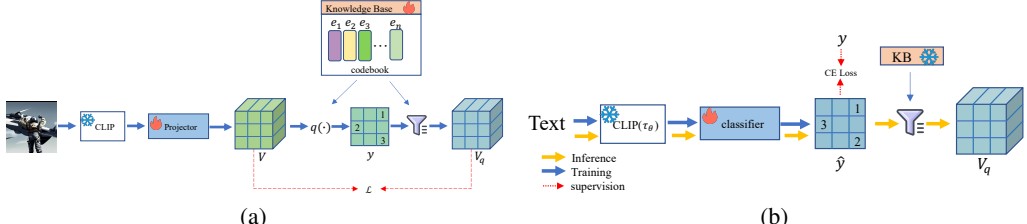

Figure 3: (a) The codebook training. $V$ is the image feature after mapping. $q(\cdot)$ denotes the nearest-neighbor mapping of $V$ in the codebook embedding space, where $y$ is the nearest neighbor index and $V_q$ is the image feature quantized by the codebook. (b) The process of classifier training. The text encoder is consistent with $\tau_\theta$ in Fig. 2. $\hat{y}$ denotes the codebook entry assignment of visual tokens predicted from text features, supervised by $y$ obtained in (a).

Knowledge Base (KB) to provide global visual semantics related to the input text during image generation, improving pose accuracy while maintaining image quality. Meanwhile, the Dynamic pose Masking (DM) mechanism adaptively adjusts the weights of pose-related regions, enabling the model to better balance local pose precision. By injecting them into the diffusion process, our framework leverages global (KB) and local (DM) control to improve HIG quality. In summary, the contributions of this paper are as follows:

- Designing a visual KB to improve pose generation accuracy while preserving image quality.

- Designing DM to control the precise generation of poses.

- Injecting KB and DM into different stages of the SD to enhance pose accuracy via global and local control without compromising image quality.

## 2 RELATED WORK

**Pose-Guided Human Iage Generation.** Previous pose-guided approaches take a source image and pose as input to generate photo-realistic human images with preserved appearance, relying on GANs/VAEs for conditional generation (Men et al., 2020; Tang et al., 2020; Ma et al., 2017; 2021; Yang et al., 2021). Recently, Bhunia et al. (2023) introduce texture diffusion modules and disentangled classifier-free guidance to accurately model appearance-pose relationships and ensure input-output consistency. Zhang et al. (2022) propose a novel Dual-task Pose Transformer Network (DPTN), which introduces an auxiliary task (i.e., source to source task) and exploits the dual-task correlation to promote the performance of PGPIG. Shen et al. (2023) presented a Progressive Conditional Diffusion Model (PCDM), which narrows the gap between character images under target poses and source poses step by step via a three-stage procedure. The above methods (Bhunia et al., 2023; Zhang et al., 2022; Shen et al., 2023) remain highly dependent on the distribution of source images, operating under the paradigm of pose transfer that necessitates original image inputs. Recent work based on pose guided T2I dispenses with the need for source images entirely: HumanSD (Ju et al., 2023b) enhances the pose accuracy of HIG via heatmap-guided losses and Stable-Pose (Wang et al., 2024) employs coarse-to-fine masking for precise HIG. GRPose (Yin et al., 2025) establish a graph topological structure between the pose priors and latent representation of diffusion models to capture the intrinsic associations between different pose parts. However, these methods prioritize pose fidelity and lack global image quality assurance, prompting us to introduce a visual KB, which enhances the pose accuracy of HIG and ensures image quality via global guidance and a DM to guarantee pose precision via local control.

**Controllable Diffusion Models.** Large-scale T2I diffusion models (Rombach et al., 2022; Ramesh et al., 2021; 2022) excel in generating diverse high-quality images but lack precision with text-only prompts. Recent works have improved controllability mainly through two approaches: training full T2I models (e.g., Composer (Huang et al., 2023) decomposes and composes diffusion models for multi-control, HumanSD (Ju et al., 2023b) fine-tunes Stable Diffusion with pose-specific losses) or developing plug-in adapters for pre-trained models (e.g., T2I-Adapter (Mou et al., 2024) and GLIGEN (Li et al., 2023) integrate lightweight adapters into frozen SD, ControlNet (Zhang et al., 2023) encodes conditions via trainable encoder copies, Uni-ControlNet (Zhao et al., 2023) enables multi-scale conditional injection, and ControlNet++ (Li et al., 2024) optimizes cycle consistency). Our approach aligns with the adapter-based paradigm, freezing the pre-trained SD model.

**Relation to VQ-VAE.** The core idea of Vector Quantised-Variational AutoEncoder (VQ-VAE) (Van Den Oord et al., 2017) is to learn discrete latent representations by mapping continuous features into a codebook of quantized vectors. This foundational framework has been used in various tasks. For example, PCTPose (Geng et al., 2023) employs a discrete codebook to model 2D human joint relations, while VQ-VAE-2 (Razavi et al., 2019) extends VQ-VAE for large-scale image generation. Both methods adopt EMA updates to alleviate codebook collapse, but such updates require additional hyperparameters (e.g., decay rates) and converge slowly. Moreover, the pixel-wise sampling in VQ-VAE-2 is time-consuming. In addition, Zero-shot T2I (Ramesh et al., 2021) explores VQ-VAE-based frameworks for text-to-image generation, and Text2Human (Jiang et al., 2022) uses a VQ-VAE-style codebook to model local clothing attributes in HIG. In contrast, during the separate KB training stage, we first discretize image features into a visual semantic codebook with an entropy loss to prevent collapse, then train a classifier to retrieve visual tokens from the codebook based on text prompts, providing efficient semantic guidance for image generation.

## 3 METHOD

### 3.1 OVERVIEW OF KB-DMGEN

Our goal is to generate high-quality human images conditioned on pose priors. To this end, we propose KB-DMGen, a Stable Diffusion based framework equipped with two adapters: a KB Adapter and a DM Adapter in Fig. 2. Specifically, SD provides the backbone, where an input image $x \in \mathbb{R}^{H \times W \times 3}$ is encoded into latent $z_0 = E(x)$, perturbed into $z_t$ by Gaussian noise, and denoised across $T$ steps. At each step, a U-Net conditioned on text $p$, pose priors $\varphi$, KB, and DM predicts the noise to reconstruct $\hat{x}$.

## 3.2 KNOWLEDGE BASE

The KB is composed of diverse codebook entries, each encoding distinct visual feature priors. The training of KB consists of two stages: first, aligning image features to the codebook vectors, and second, training a classifier to predict codebook assignments.

**Stage 1: Codebook training.** The goal of codebook training is to learn a discrete representation of image features that can be queried by text. As shown in Fig. 3a, similar to VQ-VAE (Van Den Oord et al., 2017), we freeze the CLIP encoder and add a projection layer to process the encoded features, facilitating stable training of the codebook. The codebook consists of trainable entries $\mathcal{Z} = (e_1, e_2, \ldots, e_K)$, where $\mathcal{Z} \in \mathbb{R}^{K \times C}$, with $K$ denoting the number of codebook entries and $C$ the feature dimension.

Given an image, we encode it to obtain dense image features $V = (v_1, v_2, \cdots, v_N) \in \mathbb{R}^{N \times C}$, where $N = H \times W$ is the number of spatial tokens and $C$ is the feature dimension. Each token $v_i$, $i \in \{1, 2, \cdots, N\}$, is quantized via a nearest-neighbor lookup in the codebook embedding space, as defined by the following equation:

$$q(v_i = k | V) = \begin{cases} 1 & \text{if } k = \arg\min_j \|v_i - e_j\|_2 \\ 0 & \text{otherwise,} \end{cases} \tag{1}$$

where $j \in \{1, 2, \cdots, K\}$ and $q(v_i = k | V)$ denotes a one-hot indicator. We use $q(v_i)$ to represent the index to the corresponding codebook entry. We denote the set of codebook indices for image token as $y = (q(v_1), q(v_2), \cdots, q(v_N))$, which not only allows us to select the quantized image features $V_q = (e_{y_1}, e_{y_2}, \cdots, e_{y_N}) \in \mathbb{R}^{N \times C}$, but also serves as the target labels for subsequent classifier training. Finally, the VQ-VAE loss consists of a reconstruction term and a commitment term, formulated as:

$$\mathcal{L}_{\text{VQ}} = \|\text{sg}[V_q] - V\|^2 + \beta \|V_q - \text{sg}[V]\|^2, \tag{2}$$

where sg denotes stop-gradient and $\beta = 0.25$.

After the basic VQ-VAE objective, we encourage balanced codebook usage. Without regularization, training often collapses to a few entries. To mitigate this, we add an entropy loss that promotes diverse token assignments. Formally, recall that the quantization process yields the index set of image tokens $y = (q(v_1), q(v_2), \cdots, q(v_N))$, where each $q(v_i)$ corresponds to the selected codebook entry. To measure the usage distribution of the codebook, we define the empirical frequency of token assignments as:

$$p_k = \frac{1}{N} \sum_{i=1}^{N} \mathbf{1}[q(v_i) = k], \quad p \in \mathbb{R}^K, \tag{3}$$

where $K$ denotes the number of codebook entries and $p_k$ is the average usage probability of the $k$-th entry. Based on this distribution, we compute the Shannon entropy:

$$\mathcal{H}(p) = -\sum_{k=1}^{K} p_k \log(p_k + \epsilon), \tag{4}$$

where $\epsilon$ is a small constant for numerical stability. The entropy is then normalized by the maximum entropy $\log K$, yielding a value in $[0, 1]$:

$$\mathcal{H}_{\text{norm}}(p) = \frac{\mathcal{H}(p)}{\log K}. \tag{5}$$

Finally, the entropy regularization loss is defined as:

$$\mathcal{L}_{\text{entropy}} = 1 - \mathcal{H}_{\text{norm}}(p), \tag{6}$$

which penalizes skewed token usage and encourages uniform allocation across all codebook entries. The overall codebook training objective becomes:

$$\mathcal{L} = \mathcal{L}_{\text{VQ}} + \mathcal{L}_{\text{entropy}}. \tag{7}$$

**Stage 2: Classifier training.** In this stage, the classifier is trained to use text features to predict quantized image features from learned $\mathcal{Z}$. As shown in Fig. 3b, Given an input text, we extract a

Table 1: Comparison with state-of-the-art methods on the Human-Art dataset.

| Dataset | Method | Pose Accuracy | | | Image Quality | | T2I Alignment |
|---------|--------|---------------|---|---|---------------|---|---------------|
| | | AP(%) ↑ | CAP(%) ↑ | PCE ↓ | FID ↓ | KID ↓ | CLIP(%) ↑ |
| | SD* | 0.24 | 55.71 | 2.30 | 11.53 | 3.36 | 33.33 |
| | T2I-Adapter | 27.22 | 65.65 | 1.75 | 11.92 | 2.73 | 33.27 |
| | ControlNet | 39.52 | 69.19 | 1.54 | 11.01 | 2.23 | 32.65 |
| Human-Art | Uni-ControlNet | 41.94 | 69.32 | 1.48 | 14.63 | 2.30 | 32.51 |
| | GLIGEN | 18.24 | 69.15 | 1.46 | – | – | 32.52 |
| | HumanSD | 44.57 | 69.68 | 1.37 | 10.03 | 2.70 | 32.24 |
| | GRPose | 49.50 | 70.84 | 1.43 | 13.76 | 2.53 | 32.31 |
| | Stable-Pose | 48.88 | 70.83 | 1.50 | 11.12 | 2.35 | 32.60 |
| | KB-DMGen | **53.47** | **72.33** | 1.56 | 10.54 | 2.54 | 32.43 |

Table 2: Ablation study of KB, DM, and their joint effects on Human-Art dataset.

| Components | AP (%)↑ | CAP (%)↑ | PCE↓ | FID↓ | KID↓ | CLIP(%)↑ |
|-----------|---------|----------|------|------|------|----------|
| Stable-Pose (Base) | 48.88 | 70.83 | 1.50 | 11.12 | 2.35 | 32.60 |
| +KB | 50.73 | 71.04 | 1.58 | 11.28 | 2.52 | 32.47 |
| +DM | 49.16 | 70.63 | 1.53 | 11.41 | 2.35 | 32.44 |
| +KB+DM | 51.40 | 71.17 | 1.54 | 10.56 | 2.54 | 32.41 |
| +KB w/o $\gamma_t$+DM w/o Sig. | **53.47** | **72.33** | 1.56 | 10.54 | 2.54 | 32.43 |

embedding feature $F$ using a freezed pretrained text encoder (e.g., CLIP). Next is the setting of the **classifier**. The dimension of the result after $F$ is flattened and projected is changed:

$$X = \text{Linear}(\text{flatten}(F)). \tag{8}$$

where $X \in \mathbb{R}^{N \times V}$. $N$ matches the number of image tokens, hence the same symbol and $V$ is feature dimension. subsequently, like the operation of PCTPose, four MLP-Mixer (Tolstikhin et al., 2021) blocks is used to process the features $X$, and output the logits of token classification:

$$\hat{y} = \mathcal{M}(X), \tag{9}$$

where $\hat{y}$ has the shape of $\mathbb{R}^{N \times K}$ and $K$ is the number of codebook entries. The supervision $y$ is obtained from a pretrained and frozen stage 1, which takes the input image corresponding to the current text inputs. We optimize $\hat{y}$ against $y$ using cross-entropy loss:

$$\mathcal{L}_{\text{cls}} = \text{CE}(y, \hat{y}). \tag{10}$$

This learning process enables text features to predict the corresponding visual codebook entries, allowing text-based retrieval of visual tokens.

**KB Embedding Diffusion Model.** In this stage, the pretrained KB is integrated into a diffusion model to guide the image generation process. Fig. 2 illustrates that the text $p$ is encoded by text encoder $\tau_\theta$ to query the KB and generate visual semantic priors $z_t^{''}$, which corresponds exactly to the stage-2 inference process (Fig.3b), where the feature $V_q$ is equivalent to $z_t^{''}$. To achieve the integration of $z_t^{''}$ and U-Net, we employ a FiLM-style (Perez et al., 2018) modulation block. Concretely, the feature $z_t^{''}$ retrieved from the codebook $\mathcal{Z}$ is passed through an Multi-Layer Perceptron (MLP) (LeCun et al., 2015) to generate the affine parameters $(\gamma_{cb}, \beta_{cb})$. Meanwhile, the diffusion time step $t$ embedding $\tau_t \in \mathbb{R}^{C_t}$ is projected to generate $(\gamma_t, \beta_t)$, which serves as a global modulation to further ensure overall image quality. The final modulation parameters are obtained by combining both sources:

$$\gamma = \gamma_{cb} \odot \gamma_t, \quad \beta = \beta_{cb} \odot \beta_t, \tag{11}$$

where $\odot$ means element-wise multiplication. The $z_t^{'''}$ in Fig. 2 can be obtained and injected U-Net:

$$z_t^{'''} = z_t \odot (1 + \gamma) + \beta, \tag{12}$$

where $(\gamma, \beta) = \text{MLP}(z_t^{''}, \tau_t)$ are adapted by both visual priors $z_t^{''}$ and time step $t$. Therefore, the KB-conditioned image generation $\kappa_\theta$ can be formulated as:

$$\kappa_\theta(z_t, \mathcal{Z}, t) = z_t \odot (1 + \gamma) + \beta, \tag{13}$$

where $\mathcal{Z}$ denotes the codebook set in KB.

### 3.3 DYNAMIC MASKING

Our method builds upon the coarse-to-fine Pose-Masked Self-Attention (PMSA) of Stable-Pose (Wang et al., 2024), which applies Gaussian-dilated pose masks to gradually refine latent representations. The difference is that we combine Dynamic Mask and PMSA (DM-PMSA).

**DM-PMSA.** A binary pose mask $m_k$ is obtained from the skeleton image, downsampled to match the latent feature $z_t \in \mathbb{R}^{c \times h \times w}$ (see Fig. 2), and dilated by Gaussian kernels of decreasing sizes $\{k_1 > \cdots > k_N\}$—we directly adopt the optimal Gaussian kernel configuration $\{23, 13\}$ with standard deviation $\sigma = 3$ from Stable-Pose—to produce $m_{k_1}, \ldots, m_{k_N}$. The latent feature $z_t$ is then processed by a sequence of $N = 2$ ViT blocks, each associated with one of the Gaussian-dilated masks in a coarse-to-fine manner. Within each block, the all patch embeddings of $z_t$ are projected into querie $Q$, key $K$, and value $V$, and standard attention logits (Vaswani et al., 2017) are computed as:

$$\text{dots} = \frac{QK^\top}{\sqrt{d}}, \tag{14}$$

where $d$ is the projected channel. Then, Attention is restricted to pose-relevant regions and dynamically modulated:

$$A_k = \text{softmax}\left\{(1 + \delta m_k)(\text{dots} + \text{AttnMask}(m_k))\right\}V, \tag{15}$$

where $\delta = \text{Sigmoid}\{\textbf{MLP}(t)\}$ is a timestep-dependent modulation factor applied only to pose regions via $\delta m_k$, and $\text{AttnMask}(m_k) \in \mathbb{R}^{l \times l}$ where $l = h \times w$ is a pose-aware attention mask: entries corresponding to pose-related patches ($m_k = 1$) are set to 0, while all other entries are assigned $-\infty$, suppressing attention outside pose regions. This coarse-to-fine progression of $N$ blocks gradually steers the latent representation to align with the target pose, while the dynamic modulation provides additional flexibility in controlling pose influence across timesteps.

We define $F_\theta^{\text{dyn}}$ as the DM-PMSA process, and the **conditioning function** $\nu_\theta(z_t, \varphi, t)$ as the combination of the DM-PMSA and the pose encoder:

$$\nu_\theta(z_t, \varphi, t) = F_\theta^{\text{dyn}}(z_t, \varphi, t) + \beta_\theta(\varphi), \tag{16}$$

$$z_t^{'} = z_t + \nu_\theta(z_t, \varphi, t) \tag{17}$$

where $\varphi \in \mathbb{R}^{h \times w \times 3}$ is the input pose skeleton in Fig. 2. In this way, $\nu_\theta$ captures both the spatial dependencies between body parts (via $F_\theta^{\text{dyn}}$) and global pose features (via $\beta_\theta$), with adaptive modulation across diffusion timesteps. Following Stable-Pose, $\beta_\theta$ is a trainable encoder.

As shown in Fig. 2, in our framework, DM is injected in the first layer ($L_{inj} = 1$) for fine-grained local pose control, while KB is injected in intermediate and later layers ($L_{inj} = 4, 7, 10$) to provide global visual guidance.

### 3.4 LOSS OF KB-DMGEN

As shown in Fig. 2, the denoising network $\epsilon_\theta$ adopts a UNet backbone. Let $\epsilon_\theta(\mathbf{z}_t, t, p)$, $t \in \{1, \cdots, T\}$, represent a T-step denoising UNet with gradients $\nabla\theta$ over a batch and input text prompt $p$. The denoising model predicts the noise as:

$$\epsilon_{\text{pred}} = \epsilon_\theta(z_t, t, \tau_\theta(p), \nu_\theta(z_t, \varphi, t), \kappa_\theta(z_t, \mathcal{Z}, t)), \tag{18}$$

where $\tau_\theta$ is the text encoder, $\nu_\theta(z_t, \varphi, t)$ is DM conditioning function in Sec. 3.3 and $\kappa_\theta(z_t, \mathcal{Z})$ is the KB-guided conditioning function in Sec. 3.2.

The reconstruction error outside the pose regions is computed as:

$$\mathcal{L}_{\text{um}} = \mathbb{E}_{\mathbf{z}, p, \varphi, \epsilon, \mathcal{Z} \sim \mathcal{N}(0, I), t}\left[\|(\epsilon - \epsilon_{\text{pred}}) \odot (1 - m_{k_N})\|_2^2\right], \tag{19}$$

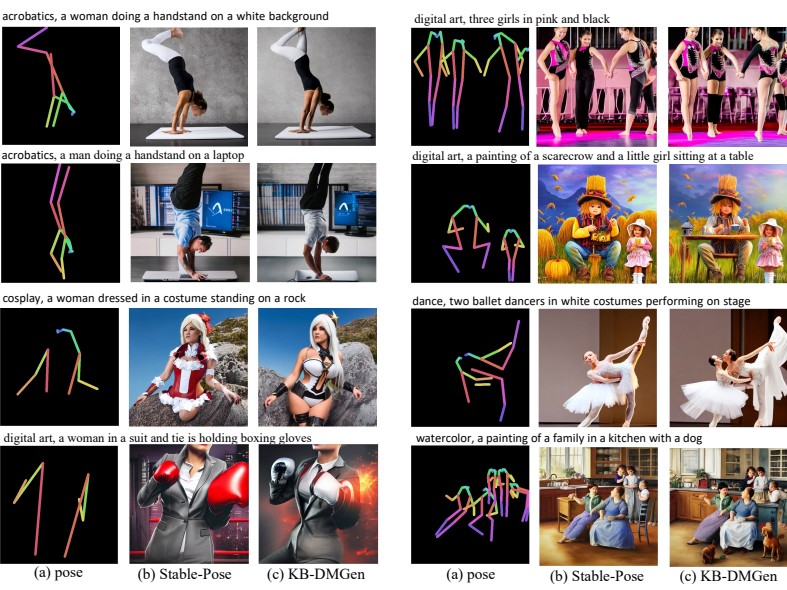

Figure 4: Comparison of visualization between Stable-Pose (Wang et al., 2024) and KB-DMGen on the Human-Art dataset.

Table 3: The KB and DM ablation experiments on the Human-Art dataset.

| Components | AP (%)↑ | CAP (%)↑ | FID↓ |
|---|---|---|---|
| Base | 48.88 | 70.83 | 11.12 |
| +KB | 50.73 | 71.04 | 11.28 |
| +KB(w/o $\mathcal{L}_{\text{entropy}}$) | 50.61 | 71.16 | 11.33 |
| +KB(w/o $\gamma_{cb}\&\beta_{cb}$) | 48.34 | 70.57 | 11.70 |
| +KB(w/o $\gamma_t\&\beta_t$) | **51.12** | **71.70** | 11.85 |
| +DM | 49.16 | 70.63 | 11.41 |
| +DM(w/o sigmoid) | 49.36 | 71.43 | 12.45 |

where $m_{k_N}$ be the finest-level pose mask and $\epsilon$ denote the Gaussian noise added to the latent encoding $z_t$ at timestep $t$. This term ensures that the model maintains consistency in the background and non-pose regions. For pose-relevant areas, the error is measured by:

$$\mathcal{L}_{\text{m}} = \mathbb{E}_{\mathbf{z},p,\varphi,\epsilon,\mathcal{Z}\sim\mathcal{N}(0,I),t}\left[\|(\epsilon - \epsilon_{\text{pred}}) \odot m_{k_N}\|_2^2\right]. \tag{20}$$

This loss explicitly enforces accurate reconstruction within the pose-constrained regions. Finally, the overall training objective is a weighted sum:

$$\mathcal{L} = \mathcal{L}_{\text{um}} + \alpha\mathcal{L}_{\text{m}}, \tag{21}$$

where $\alpha = 5$ is a hyperparameter emphasizing the masked (pose) regions, following the optimal design of Stable-Pose.

## 4 EXPERIMENTS

### 4.1 EXPERIMENTAL SETTINGS

**Datasets** We evaluated our model on the Human-Art (Ju et al., 2023a). The Human-Art dataset comprises 50,000 high-quality images from 5 real-world and 15 virtual scenarios, featuring human bounding boxes, key points and textual descriptions.

**Implementation Details** Similar to previous work (Wang et al., 2024), we fine-tune our model on SD with version 1.5. We utilize Adam (Kingma & Ba, 2014) optimizer with a learning rate of

Table 4: Number of parameters of FiLM across different inject layers $L_{inj}$ in KB-DMGen.

| $L_{inj}$ | $\{4\}$ | $\{7\}$ | $\{10\}$ | all$\{4, 7, 10\}$ |
|---|---|---|---|---|
| Params (M) | 10.8 | 28.2 | 28.2 | 67.3 |

$1 \times 10^5$. We also follow (Zhang et al., 2023) to randomly replace text prompts as empty strings at a probability of 0.5, which aims to strengthen the control of the pose input. During inference, no text prompts are removed and a DDIM sampler (Song et al., 2020) with 50 time steps is utilized to generate images. We train our models with 10 epochs on all datasets. The number of codebook entried $K$ is set 1024. The batch size of inference is same as Stable-Pose (Wang et al., 2024). For the Human-Art dataset, the training is executed using five NVIDIA A6000 GPUs with 1 batch size.

For the training of KB, we fine-tune on the OpenCLIP (Cherti et al., 2023) implementation using the pretrained CLIP model (ViT-L/14) with frozen encoders. Training is performed in two stage—codebook training and classifier training—both using the same strategy over 30 epochs with Adam (Kingma & Ba, 2014) (initial learning rate 0.001), a cosine decay schedule and 256 batch size on per GPU. The training and validation splits as same as Human-Art. On the Human-Art dataset, two-stage training takes about 2.5 hours with two NVIDIA 3090 GPUs.

**Metrics.** To evaluate pose accuracy, we use mean Average Precision (AP), Pose Cosine Similarity-based AP (CAP), and People Counting Error (PCE) (Cheong et al., 2022), computed using HigherHRNet (Cheng et al., 2020) to compare ground-truth poses with those extracted from generated images. For image quality assessment, we employ Fréchet Inception Distance (FID) (Heusel et al., 2017) and Kernel Inception Distance (KID) (Bińkowski et al., 2018) to measure diversity and fidelity. KID is multiplied by 100 for Human-Art.Text-image alignment is evaluated using the CLIP's score (Radford et al., 2021).

## 4.2 COMPARISON WITH SOTA METHODS

Our method with other state-of-the-art (SOTA) approaches are shown in Table 1.

**On the Human-Art dataset**, our final model achieves the highest AP and CAP, reaching 53.47 and 72.33 respectively, which surpasses GRPose (Yin et al., 2025) by +3.97 AP and +1.49 CAP. Compared with the baseline Stable-Pose (Wang et al., 2024), the improvement is even more substantial (+4.59 AP and +1.50 CAP). Meanwhile, our method maintains competitive global image quality, with FID and KID comparable to other strong methods. This demonstrates that our design not only significantly improves pose accuracy but also preserves overall image quality. Qualitative results in Fig. 4 further confirm that our method produces visually more faithful images with superior pose accuracy and semantic consistency. What's more, more visualization results including comparison of results of each module will be presented in detail in Appendix (A.5)

## 4.3 ABLATION STUDIES

Our method builds on Stable-Pose as the baseline, with ablations on the Human-Art dataset.

**Overall Effects of KB and DM.** We first evaluate the effectiveness of the proposed KB and DM modules, as well as their joint impact on performance. As shown in Table 2, adding KB improves both AP and CAP ensuring image quality. It outperforms the recent SOTA method GRPose (Yin et al., 2025) with a 1.23 AP and 0.2 CAP improvement, while achieving better image quality with a 2.48 drop in FID and a slight 0.01 decrease in KID. DM alone mainly enhances AP slightly. Importantly, their combination consistently improves on AP, CAP, FID, while other indicators have only slightly worsened, demonstrating the complementarity between KB and DM. Surprisingly, when removing the temporal scaling factor $\gamma_t \& \beta_t$ from KB in Eq. 11 and the sigmoid gating from DM in Eq. 15, compared with KB+DM, this method achieves the best AP and CAP with only a slight increase in PCE, demonstrating the better complementary role of KB (w/o $\gamma_t \& \beta_t$) global guidance and DM (w/o Sig.) local refinement, as further confirmed by the visualizations in Appendix (A.4). A detailed discussion on decomposing text and extracting corresponding visual features from the KB is provided in Appendix (A.1).

**Effect of KB.** To further investigate the KB design, we conduct several ablations as reported in Table 3. Removing the entropy regularization $\mathcal{L}_{\text{entropy}}$ degrades performance in AP and FID. Eliminating the affine modulation parameters $(\gamma_{cb}, \beta_{cb})$ causes a significant drop in AP and CAP, highlighting KB necessity for HIG. Interestingly, removing the timestep-dependent modulation $(\gamma_t, \beta_t)$ yields the highest CAP and AP, but comes at the cost of worse generative fidelity (FID). We omit KID for brevity, as its variance across settings is negligible. The impact of codebook size $K$ on the index results will be discussed in the appendix (A.3).

**Effect of DM.** Table 3 also summarizes the ablations on the DM module. While DM improves AP compared to the baseline, removing the sigmoid gating ($\delta = \text{Sigmoid}\{\textbf{MLP}(t)\}$) in Eq. 15—where the MLP has only 0.6M parameters—further increases AP and CAP but causes severe degradation of FID by about 1. This indicates that the Sigmoid gate is crucial for stabilizing the modulation strength and preserving global quality, even though it slightly limits precision gains.

**Joint Effects of $\gamma_t \& \beta_t$ and Sigmoid.** The joint analysis in Table 2 shows that the combination of KB without $\gamma_t \& \beta_t$ and DM without sigmoid achieves the best AP (53.47) and CAP (72.33). When used individually, each component requires additional constraints to stabilize training, whereas their joint application is complementary, with KB providing global control and DM providing local refinement, achieving the best balance between pose accuracy and image quality.

### 4.4 PARAMETERS ANALYSIS

Table 4 lists the trainable parameters of the FiLM modules (see Fig. 2) for different injected layers $L_{inj}$ in KB-DMGen. Overall, KB-DMGen has 106M more trainable parameters than Stable-Pose and 58M more than GRPose, most of which come from the FiLM modules, with a small portion from other mapping layers. Detailed KB parameter statistics at different training stages are reported in Appendix A.2.

## 5 CONCLUSION

We propose an image generation method combining a visual KB with DM, validated by experiments to generate high-quality images. Future work includes expanding the knowledge base and further optimizing dynamic masking.

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

# A APPENDIX

## A.1 D&C ANALYSIS

As shown in Table 5, applying component-wise KB querying (+KB+DM+D&C) achieves the best overall performance compared with KB+DM. Specifically, D&C leverages the simple and regular structure of HumanArt descriptions to decompose each text into three parts—type, object, and status (e.g., "cartoon, an animal character with a sword in the woods" -> type = "cartoon"; object = "an animal character"; status = "with a sword in the woods".). Each component independently queries the KB, and the results are combined into the diffusion model, which enhances semantic parsing by capturing fine-grained correspondences between visual features in KB and textual components. Fig. 7 further shows that using D&C can improve the quality of generated images and pose accuracy. Decomposing the text into multiple queries yields several results concatenated along channels, leading to about 18M more trainable parameters in KB+DM+D&C than in KB+DM.

Table 5: Ablation study of KB, DM, and their joint effects on Human-Art dataset.

| Components | AP (%)↑ | CAP (%)↑ | PCE↓ | FID↓ | KID↓ | CLIP(%)↑ |
|---|---|---|---|---|---|---|
| Stable-Pose (Base) | 48.88 | 70.83 | 1.50 | 11.12 | 2.35 | 32.60 |
| +KB+DM | 51.40 | 71.17 | 1.54 | 10.56 | 2.54 | 32.41 |
| +KB+DM+D&C | **51.71** | **71.40** | 1.53 | 10.29 | 2.45 | 32.45 |

Table 6: Number of trainable parameters across different stages and the number of codebook entries $K$ in KB.

| Stage | Trainable Parameters (M) | | | |
|---|---|---|---|---|
| | $K = 256$ | $K = 512$ | $K = 1024$ | $K = 2048$ |
| 1 | 9.70 | 9.96 | 10.49 | 11.54 |
| 2 | 78.33 | 78.35 | 78.38 | 78.45 |

Table 7: The number of codebook entries $K$ Ablation experiments on Human-Art dataset

| $K$ | AP (%)↑ | CAP (%)↑ | PCE↓ | FID↓ | KID↓ | CLIP-score(%)↑ |
|---|---|---|---|---|---|---|
| 256 | 50.09 | 71.06 | 1.56 | 11.34 | 2.47 | 32.48 |
| 512 | 50.35 | 70.85 | 1.56 | 11.11 | 2.44 | 32.45 |
| 1024 | **50.73** | 71.04 | 1.58 | 11.28 | 2.52 | 32.47 |
| 2048 | 50.40 | 71.08 | 1.52 | 11.49 | 2.53 | 32.47 |

## A.2 TRAINABLE PARAMETERS OF THE KB

Table 6 reports the number of trainable parameters when training the KB alone, across its two stages and for different codebook sizes $K$. Increasing $K$ slightly increases the parameters in Stage 1, while Stage 2 remains largely unaffected.

## A.3 EFFECT OF CODEBOOK SIZE IN KB

**Codebook Size.** We analyze the impact of different codebook sizes $K$ in Table 7 using KB model alone. We observe that the performance first improves as $K$ increases, peaking at $K = 1024$, and then slightly decreases with larger codebooks. This indicates that too small a codebook limits the representational capacity, while an excessively large one introduces redundancy and instability. Thus, $K = 1024$ provides the best trade-off between efficiency and expressiveness.

## A.4 VISUALIZATION OF INJECTION BEHAVIORS

**Visualization of Codebook Injection.** In Fig. 5, we compare the effects of KB and DM on the FiLM parameters $\gamma$ and $\beta$ at different injection layers. KB alone provides strong global trends but shows large fluctuations during sampling. Introducing DM smooths the variations of $\gamma$ and $\beta$,

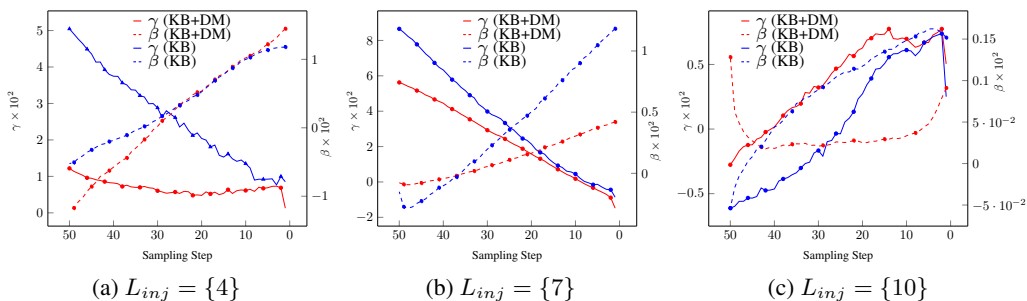

(a) $L_{inj} = \{4\}$    (b) $L_{inj} = \{7\}$    (c) $L_{inj} = \{10\}$

Figure 5: Visualization of the modulation parameters $\gamma$ and $\beta$ in Eq. 12 across the reverse sampling steps ($t = 50 \to 1$) at different injection layers ($L_{\text{inj}} = \{4, 7, 10\}$).

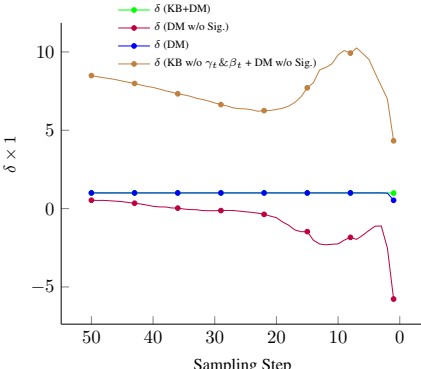

Figure 6: Visualization of the DM modulation parameter $\delta$ in Eq.15 along the reverse sampling steps ($50 \to 1$).

enabling fine-grained local refinement. Different layers exhibit distinct roles: shallow and middle layers use larger modulation to guide coarse structure generation, while deeper layers apply smaller, more stable adjustments to refine details.

**Visualization of Dynamic Mask.** As shown in Fig. 6, the curve of DM w/o Sig. gradually transitions from positive to negative during the sampling process, indicating a reversal in the modulation direction of $\delta$ in Eq. 15. This leads to unstable attention on the pose regions and results in uneven modulation. DM effectively enhances and stabilizes the magnitude of $\delta$ in DM and KB+DM. What's more, guided by the knowledge base, KB (w/o $\gamma_t \& \beta_t$)+DM (w/o Sig.) effectively enhances $\delta$'s magnitude, significantly improving generation accuracy while ensuring overall quality and enabling the dynamic mask to stay focused on key regions even without sigmoid.

### A.5 EFFECT ON GENERATED RESULTS

**Complementary analysis.** We further evaluate the effects of different injection strategies on the generated images (Fig. 8). In this figure, KB (w/o $\gamma_t \& \beta_t$)+DM (w/o Sig.) combination demonstrates a clear synergistic effect, producing more accurate poses and higher-quality images than either component individually.

**Single component analysis.** The comparison of KB and KB (w/o $\gamma_t \& \beta_t$) visualization results are shown in Fig. 9, which clearly shows that KB has an advantage in terms of generation quality. The comparison of DM and DM (w/o Sig.) visualization results results are shown in Fig. 10, which clearly shows that DM has an advantage in terms of generation quality.

**The impact of different KB and DM combinations.** We compare the visualization results of the KB+DM combination and the KB(w/o $\gamma_t \& \beta_t$)+DM(w/o Sig.) combination. As shown in Fig. 11, the result surface has advantages in both the accuracy and quality of the generated image pose.

acrobatics, a woman doing a handstand on a wooden floor

cosplay, a woman in a blue dress is laying on the ground

cosplay, a woman with white hair and white armor is holding a sword

digital art, a painting of a cat and a little girl in the grass

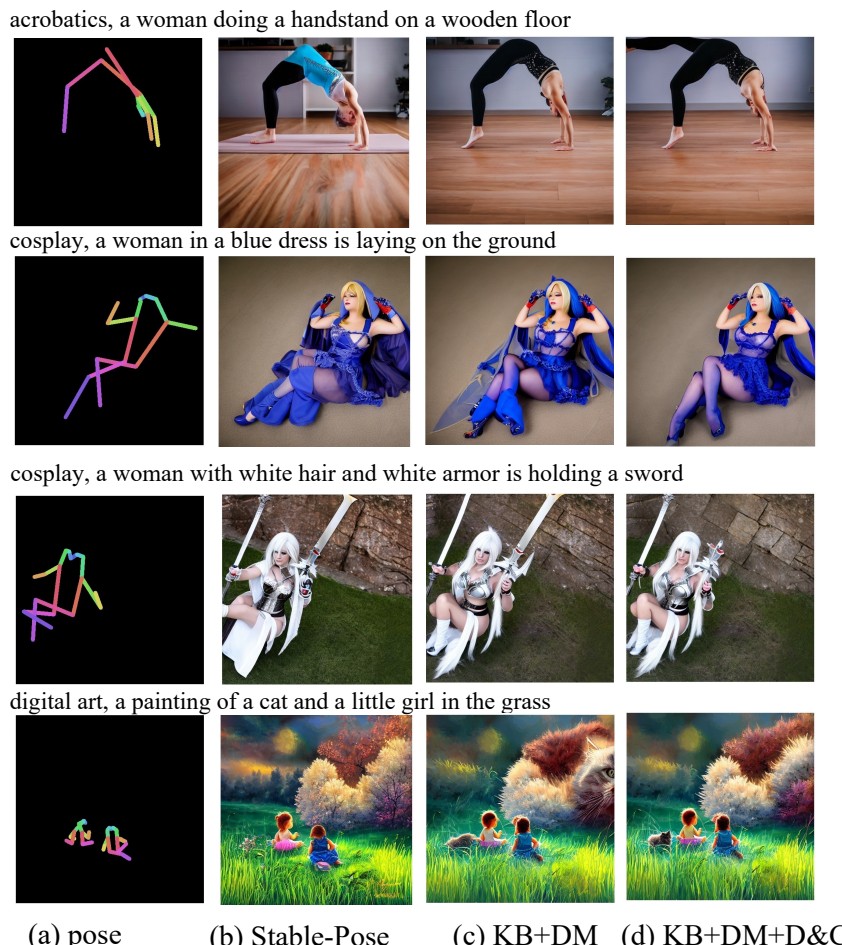

(a) pose      (b) Stable-Pose      (c) KB+DM    (d) KB+DM+D&C

Figure 7: Comparison of visualization results of KB+DM and KB+DM+D&C on the Human-Art dataset.

**Final result visualizations of KB-DMGen.** Fig. 12, 13 and 14 show more visualization results of KB-DMGen.

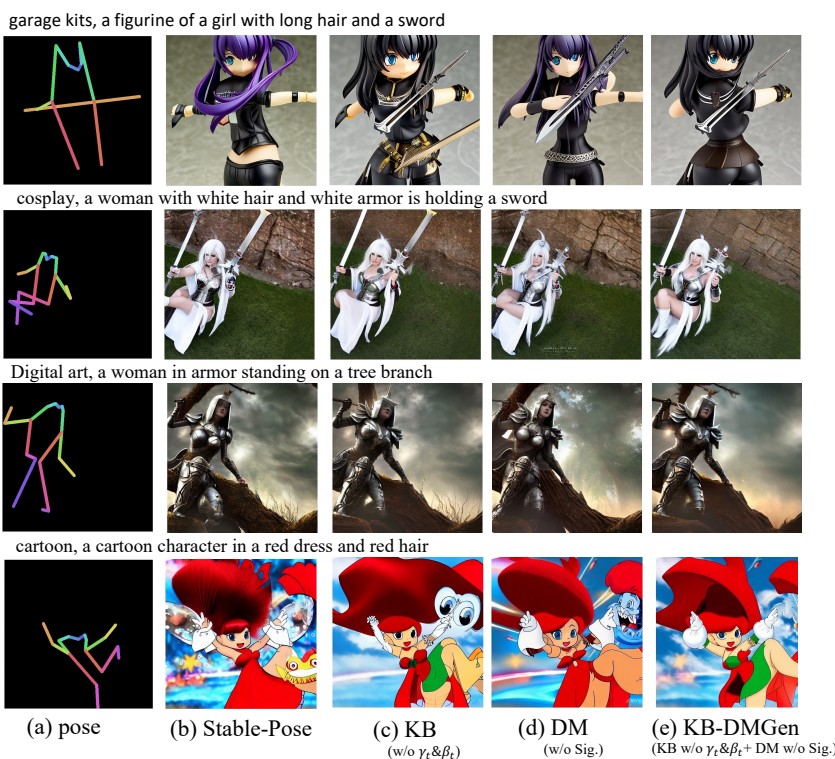

Figure 8: Visual comparison on the Human-Art dataset. The KB (w/o $\gamma_t \& \beta_t$)+DM (w/o Sig.) combination exhibits a synergistic effect, enhancing both pose accuracy and image quality beyond individual contributions. Sig. means Sigmoid.

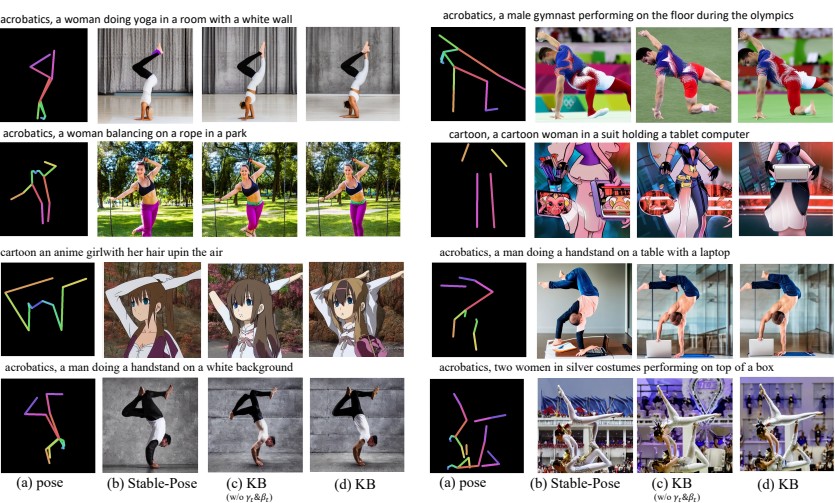

Figure 9: Visual comparison between KB (w/o $\gamma_t \& \beta_t$) and KB. The results show that the image quality is better when KB is used alone.

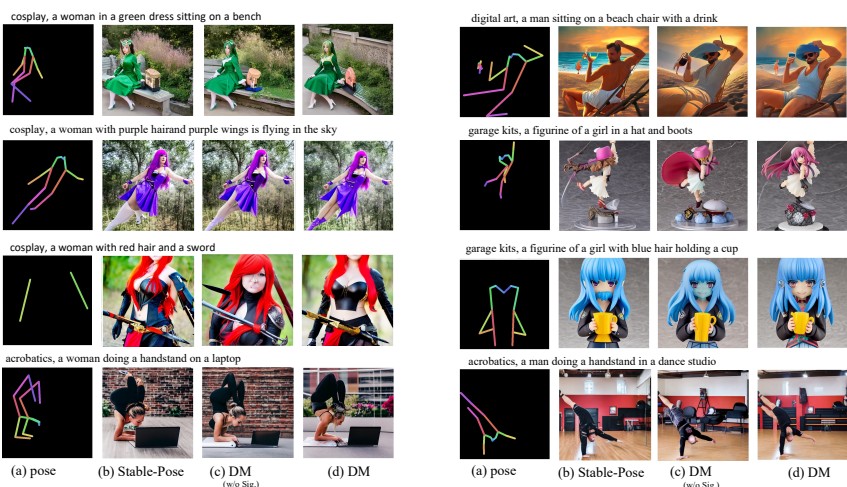

Figure 10: Visual comparison between DM (w/o Sig.) and DM on the Human-Art dataset. The results show that the image quality is better when DM is used alone.

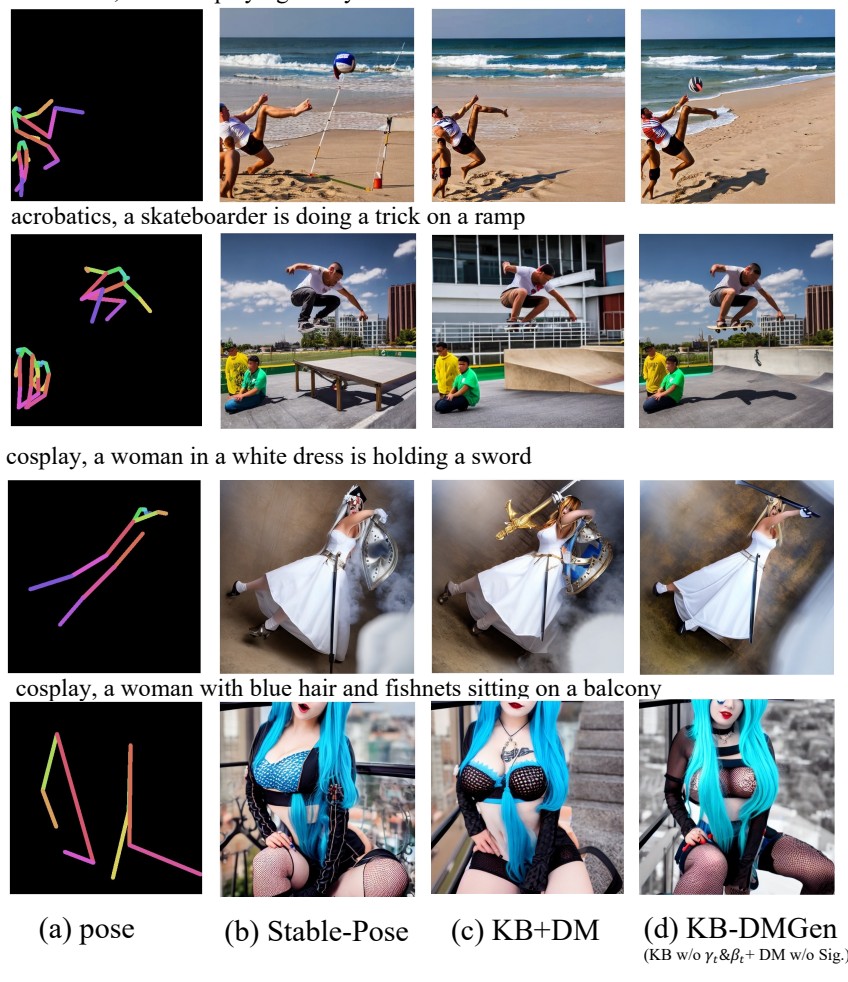

Figure 11: Visual comparison between KB+DM and KB-DMGen on the Human-Art dataset. The results show that the image quality is better.

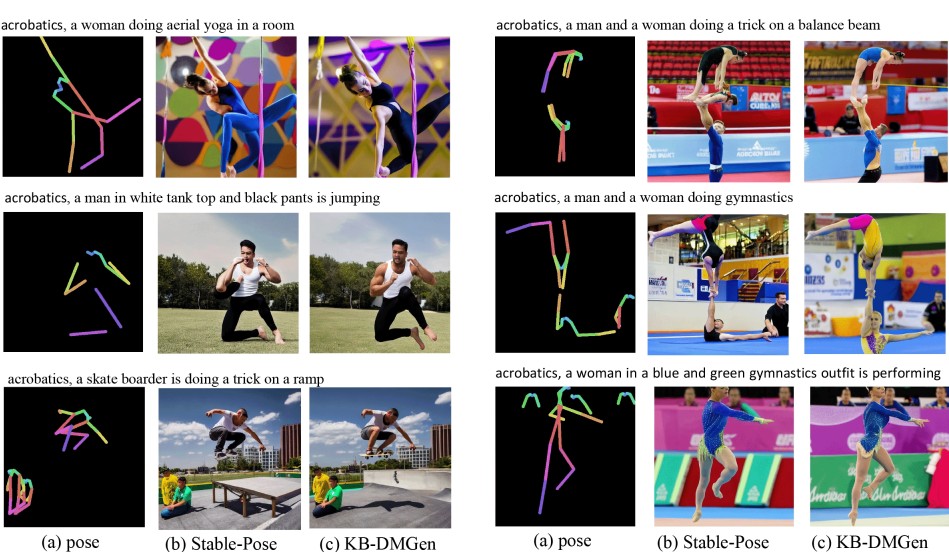

Figure 12: KB-DMGen visualization results on the Human-Art dataset.

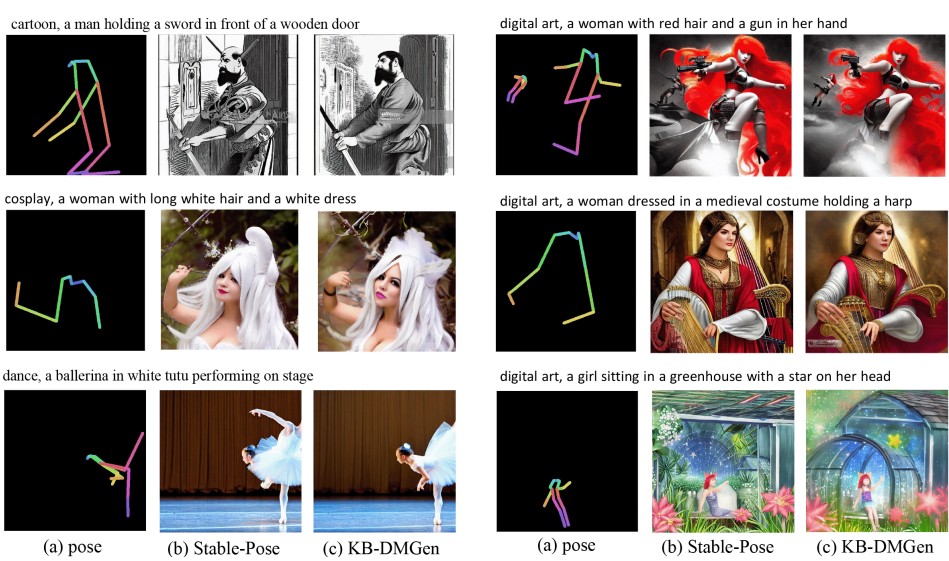

Figure 13: KB-DMGen visualization results on the Human-Art dataset.

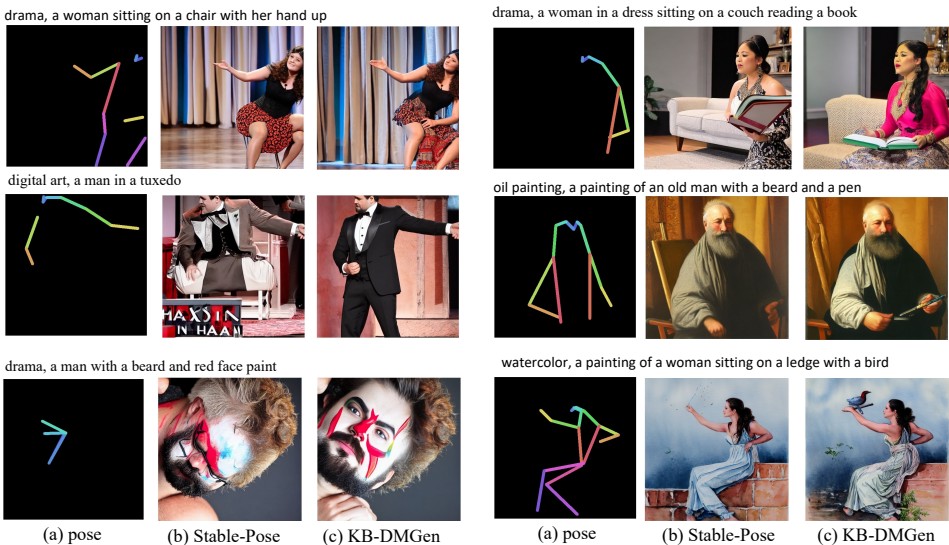

Figure 14: KB-DMGen visualization results on the Human-Art dataset.

