# OpenReview forum: "KB-DMGen: Knowlege-Based Global Guidance and Dynamic Pose Masking for Human Image Generation"
_ICLR.cc/2026/Conference — ICLR 2026 Conference Withdrawn Submission_

### Official Review · Reviewer_MPVA · 2025-10-28

**Soundness:** 2
**Presentation:** 2
**Contribution:** 1
**Rating:** 2
**Confidence:** 4

**Summary:**

This paper addresses the challenge of pose-guided human image generation using diffusion models. The authors identify a key problem in existing methods: a trade-off where improving pose accuracy often comes at the expense of overall image quality and semantic coherence. To tackle this, they propose KB-DMGen, a dual-component framework built on top of Stable Diffusion. The first component is a Knowledge Base (KB), implemented as a VQ-VAE-style visual codebook, which is queried by text prompts to provide global, semantic guidance to the diffusion process. The second component is Dynamic pose Masking (DM), an enhancement of prior masking techniques that provides fine-grained, local control over pose generation. By injecting the global KB guidance into the later stages of the UNet and the local DM control into the early stages, the method aims to achieve high pose fidelity without compromising image quality. The authors validate their approach on the HumanArt dataset, demonstrating state-of-the-art performance in pose accuracy metrics.

**Strengths:**

The paper effectively identifies a significant and practical problem in the field of controllable image generation—the inherent tension between strong conditional control (like pose) and the preservation of global image quality and semantic coherence. This provides a strong motivation for the work.

**Weaknesses:**

1 Figure 1&2 share similar meanings.

2 The proposed Knowledge Base is essentially a ``clip+VQVAE’’ structure, why is it better than a normal text encoder or a clip encoder?

3 I can not find much difference between the proposed Dynamic pose Masking and commonly used pose condition in other works.

4 There is no experiment about the length of the codebook, which makes it hard to verify the effectiveness of the proposed Knowledge Base.

5 From Table3, the improvements from the proposed method is limited.

**Questions:**

None

---

### Official Review · Reviewer_iBX4 · 2025-10-30

**Soundness:** 2
**Presentation:** 2
**Contribution:** 1
**Rating:** 2
**Confidence:** 3

**Summary:**

The authors address a key limitation in Human Image Generation (HIG) where existing methods often sacrifice overall image quality to achieve precise pose accuracy. They propose KB-DMGen, a novel framework that balances these two objectives. The method introduces a dual-control system: 1) a "Knowledge-Based Global Guidance" (KB), implemented as a visual codebook, which provides coarse, global guidance to maintain visual coherence while steering the pose, and 2) "Dynamic pose Masking" (DM), which provides fine-grained, local control to enhance pose precision. By integrating these global and local controls at different diffusion stages, the framework reportedly improves pose accuracy without compromising image quality, setting a new state-of-the-art on the HumanArt dataset.

**Strengths:**

- The proposed architecture (KB-DMGen) is intuitive and novel. The separation of control into a coarse, global guidance (KB) for overall quality and a fine-grained, local control (DM) for pose precision is a sensible and well-motivated design.
- The claim of achieving new state-of-the-art performance on the HumanArt dataset, measured by both AP and CAP metrics, provides strong empirical validation for the method's effectiveness.

**Weaknesses:**

- Dependency on Outdated Architecture: The method's foundation on Stable Diffusion 1.5 is a significant concern. As this backbone is now considerably outdated, the paper's impact is limited unless the authors demonstrate that the proposed KB-DMGen modules are portable to newer, state-of-the-art architectures (e.g., Diffusion Transformers).
- Sub-optimal Photorealism: The qualitative results, while demonstrating accurate poses, appear to lack the high-fidelity photorealism characteristic of current SOTA generative models. This quality ceiling, likely inherited from the SD1.5 base, compromises the practical appeal of the generated images.
- Insufficient Comparative Evaluation: The visual evaluation is not comprehensive. A stronger case would be made by including direct comparisons against other recent, relevant works (such as OmniControl) and benchmarking against competitive commercial models (like Nanobanana) to fully contextualize the method's performance.

**Questions:**

See Weaknesses.

---

### Official Review · Reviewer_YQds · 2025-10-31

**Soundness:** 2
**Presentation:** 2
**Contribution:** 3
**Rating:** 4
**Confidence:** 3

**Summary:**

This paper proposes KB-DMGen, simultaneously introducing global semantic guidance of the visual knowledge base and local fine-grained control of dynamic pose masks in the Stable Diffusion framework; Both are injected at different layers and moments to strike a balance between pose alignment and overall image quality.

**Strengths:**

The problems of the existing Pose-guided T2I are pointed out. Based on this, the global semantics and local poses are decoupled and injected, and the structure is intuitive.

This article proposes a two-stage training and incorporates entropy regularization to prevent codebook collapse, and then modulates U-Net in a FiLM manner.


Add a time-step adaptive modulation factor to the Gaussian dilation mask of Stably Pose to achieve "coarse-to-fine" pose constraints.

**Weaknesses:**

KB relies on CLIP-L/14 features and is trained/validated on Human-Art; There is a lack of robustness assessment for cross-dataset or out-of-domain text retrieval (such as switching to the OpenCLIP model or changing the description style).

In addition to AP/CAP/PCE, the robust segmentation index of human keypoint visibility/occlusion and a more fine-grained measurement of text consistency can be added. Currently, the CLIP-score difference is extremely small.

**Questions:**

The article points out that FiLM injection brings significant additional parameters, but the latency and video memory of training and inference seem not to have been quantified.

---

### Official Review · Reviewer_GXwB · 2025-11-01

**Soundness:** 3
**Presentation:** 2
**Contribution:** 3
**Rating:** 4
**Confidence:** 3

**Summary:**

The submission proposes KB-DMGen for pose-guided human image generation, combining a text-queried visual codebook (KB) that provides global semantic guidance via a two-stage VQ-style training with entropy regularization and FiLM-based late-layer modulation, with a Dynamic Masking (DM) module that extends pose-masked self-attention using timestep-dependent modulation for fine-grained pose control; integrated into a frozen Stable Diffusion backbone (early-layer DM, mid/late-layer KB), the method achieves SOTA AP/CAP on Human-Art while maintaining competitive FID/KID, with extensive ablations (entropy, FiLM variants, sigmoid gating, codebook size, injection layers, and optional text Decompose & Compose) supporting the design and clarifying stability–accuracy trade-offs.

**Strengths:**

(1) Clear, well-motivated split of global (KB) and local (DM) control; sensible layer-wise injection via FiLM.

(2) Practical adapter-based design; modest KB training cost; strong improvements in pose metrics.

(3) Comprehensive ablations (entropy, FiLM params, sigmoid gating, codebook size, injection layers, D&C) that illuminate stability–accuracy trade-offs.

**Weaknesses:**

(1) Limited datasets:

All core results are on Human-Art. This is a specialized dataset (real + virtual). Lack of evaluation on broader human-centric datasets (e.g., DeepFashion, Human3.6M variants, or more diverse captions) limits generality claims.

(2) KB training supervision coupling:

Stage-2 classifier supervision y is derived from Stage-1 with the “corresponding image to the current text.” This implies access to paired text-image data with good alignment. More clarity is needed on noise in text-image pairs and robustness when captions are imperfect (common in the wild).

(3) Ambiguity and novelty boundary:

The KB resembles VQ-VAE/VQGAN semantic codebooks and text-to-code retrieval used in prior T2I works; the novelty is in leveraging it as a global control stream in diffusion with FiLM and co-design with DM. This is a solid design contribution but could be seen as incremental.

(4) Stability vs. peak metrics:

The best AP/CAP settings remove stabilizing components (γt/βt and sigmoid) and slightly worsen FID/KID. While the paper is transparent about this, more discussion on safe defaults for production settings would help.

(5) Quantitative trade-offs:

In Table 1, KB-DMGen has slightly worse PCE and KID compared to some baselines in places. The paper positions these as “competitive,” but a clearer analysis of when/why PCE increases would be useful (e.g., multi-person cases? crowded scenes?).

(6) Efficiency and memory:

FiLM across {4,7,10} adds 67.3M parameters; overall +106M vs Stable-Pose. Inference-time overhead and throughput are not reported. How much slowdown vs baseline? Any impact on VRAM?

(7) D&C parsing:

The D&C relies on “simple and regular” Human-Art captions and handcrafted decomposition into type/object/status. This might not transfer to messier real-world prompts. No automatic/learned parsing baseline is provided.

**Questions:**

(1) Generalization and datasets:

Can you report results on another human-centric dataset (e.g., DeepFashion, COCO-person subset with pseudo-poses, or Human3.6M derivatives) to demonstrate robustness?

(2) Text-image alignment and robustness:

How does KB retrieval behave when captions are noisy, partial, or contain out-of-domain terms? Any analysis of failure cases in retrieval (e.g., wrong codebook tokens leading to semantic drift)?

(3) Inference cost:

What is the runtime and memory overhead during inference vs Stable-Pose and GRPose for 512×512 and 768×768? Any batched throughput metrics?

(4) KB capacity and domain shift:

How sensitive is performance to codebook domain bias? If KB is trained on Human-Art-alike images only, does it hurt transfer? Would a larger, more diverse KB (e.g., LAION-based) help?

(5) DM dynamics:

Can you visualize attention maps for DM-PMSA across timesteps to confirm increased focus on pose regions? You show δ curves; showing spatial attention changes could strengthen claims.

(6) Safe default configuration:

Which configuration do you recommend in practice: with γt/βt and sigmoid (better FID) or without (better AP/CAP)? Can you provide an adaptive schedule that interpolates for a balanced trade-off?

(7) D&C parsing:

How is D&C implemented concretely? Is it rule-based or learned? How does it handle prompts without clear type/object/status segmentation? Could you compare to a learned text decomposition (e.g., slot attention over text) baseline?

(8) Multi-person scenarios and PCE:

Where does PCE degrade? Do multi-person prompts stress KB or DM? Any targeted ablation on multi-person vs single-person subsets?

---

### Note · Authors · 2025-11-19

I have read and agree with the venue's withdrawal policy on behalf of myself and my co-authors.